# Developmental Dysplasia of the Hip: How Many Risk Factors Are Needed?

**DOI:** 10.3390/children10060968

**Published:** 2023-05-30

**Authors:** Adelina Ionescu, Mihai-Codrut Dragomirescu, Alexandru Herdea, Alexandru Ulici

**Affiliations:** 111th Department of Pediatric Orthopedics, “Carol Davila” University of Medicine and Pharmacy, Bd. Eroii Sanitari nr. 8, 050474 Bucharest, Romania; alexandru.ulici@umfcd.ro; 2Pediatric Orthopedics Department, “Grigore Alexandrescu” Children’s Emergency Hospital, 011743 Bucharest, Romania; adelina_ionescu@spitalulgrigorealexandrescu.ro (A.I.); mcodrutd@gmail.com (M.-C.D.)

**Keywords:** developmental dysplasia of the hip, DDH screening, hip ultrasound

## Abstract

Developmental dysplasia of the hip (DDH) is a progressive condition that lacks clear diagnostic and management protocols, due to insufficient data. While some advocate for universal screening, others recommend using risk factors as landmarks. In this study, we aimed to assess the risk factors associated with DDH incidence among a large population. We conducted a retrospective single-center multifactorial study between January 2019 and March 2022, including 3720 children who were investigated anamnestically, clinically, and through an ultrasound scan. We classified them into two groups: the control group with 3300 healthy children and the study group with 420 newborns diagnosed with DDH. Our analysis identified several risk factors associated with DDH, including gender, prematurity, non-vertex birth presentation, oligohydramnios, gestational diabetes, maternal hypertension, family history, associated deformities, and swaddling. We found that every DDH patient had at least two risk factors. Based on our findings, we recommend that children who present two or more risk factors for DDH be mandatorily evaluated sonographically, as well as children with clinical signs. DDH screening is recommended for each newborn for the long-term benefits of early detection and treatment.

## 1. Introduction

Developmental dysplasia of the hip (DDH) is a disease ranging from dysplasia to the actual dislocation of the hip. It is usually discovered in infancy or early childhood. Early diagnosis and treatment lead to good functional results [1].

The etiology of DDH is still unknown. A combination of genetic, environmental, and mechanical factors is blamed. The up-to-date literature mentions the following risk factors: first-born child, female sex, breech positioning in the third trimester, infant swaddling, post-maturity, factors causing limited in-utero space, large for gestational age, and family history [1,2,3]. The most significant risk factor for DDH is considered the breech position in the third trimester [2]. Genetic factors and the influence of some cytokines have also been followed up in several studies and linked to the occurrence of the disease [4,5].

Clinical examination and ultrasound of the hip are mandatory in any newborn screened for DD [6]. Universal screening for DDH is still controversial because most unstable hips recover with no treatment [7]. Some authors recommend evaluating by ultrasonography any infant with risk factors at six weeks to allow laxity and physiologic immaturity to resolute [1].

Graf has established the criteria for a normal hip on static ultrasonographic images as follows: a normal alpha angle value of over 60° between the bony acetabulum and the ilium and a beta angle less than 55° measured between the labrum and the turning point from convexity to concavity on the bony rim [8].

The goal of early treatment is to assure normal development of the hip joint and prevention of late complications. Therefore, there is a need for high suspicion and routine surveillance of DDH in infants.

In North America ultrasound screening is carried out selectively in infants with risk factors such as breech presentation, family history, and clinical hip instability [9].

The decisive role of the family history and the breech presentation in the etiology of developmental dysplasia of the hip is well known. Our retrospective study aims to see how many risk factors are needed to correlate with DDH and the degree of dysplasia, according to the Graf method.

## 2. Materials and Methods

### 2.1. Study Design

The study was conducted at the Pediatric Orthopedics Outpatient Clinic of “Grigore Alexandrescu” Children’s Emergency Clinical Hospital in Bucharest, Romania. This hospital serves as a major medical center for children in the urban area and its surroundings. The study was approved by the hospital ethics committee on 12 December 2022, and the identification number of the survey is 8. Written consent from the parents or legal guardians of all participating patients was obtained. The study was conducted between January 2019 and March 2022. A patient flow diagram depicting the study design and participant selection process is shown in Figure 1.

### 2.2. Participants

A retrospective single-center multifactorial study was conducted between January 2019 and March 2022. The study included children aged less than 4 months old, referred to the clinic for pediatric orthopedic examination and ultrasonography of the hip. All the patients were referred for examinations either as a standard guideline for all newborn babies in Romania with at least one risk factor for DDH, carried out in the first 4 months of life, or because of reported clinical signs observed by the neonatologist of the parents.

Inclusion criteria were age group of fewer than 4 months old, written consent of the parent, and lack of abnormalities such as genetic syndromes or other congenital diseases.

Exclusion criteria were age more than 4 months old, incomplete clinical data, syndromic dysplasia of the hip, or other neuromuscular diseases.

### 2.3. Study Procedure

After clinical examination and patient anamnesis regarding pregnancy and delivery, an ultrasound of both hips was carried out. The ultrasound was performed by having the child placed on one side in a special cradle, and measurements were made on the same type of device, using a linear array probe, 7.0 MHz frequency, 270° image rotation, as seen in Figure 2.

The screening was carried out by pediatric orthopedic physicians with training in Graf ultrasound hip screening for the developmental dysplasia of the hip. The measurement technique used was that according to the Graf criteria, and was used to determine the type of DDH [9], as seen in Figure 3.

### 2.4. Statistical Analysis

Data were collected and stored in the institutional informatics system. We used IBM^®^ SPSS^®^ Statistics (Version 26) and Microsoft Excel Office 2016 (Microsoft, Redmond, Washington, DC, USA). Data included categorical qualitative data (gender, fetal position, type of birth, prematurity, oligohydramnios, multiple births, gestational diabetes, maternal hypertension, family history, associated deformities, and swaddling) and continuous quantitative data (age at evaluation, gestational age, weight at birth). Descriptive tests such as frequencies, sex incidence, type of DDH, odds ratio, chi-square tests, and *p*-value were conducted. We considered a 95% confidence interval. The results were considered significantly statistical if the *p*-value was less than 0.05.

## 3. Results

The study included 3720 children, aged between 4 weeks and 4 months of life. A total of 53.76% (2000 patients) were females and 46.24% (1720 patients) were males. According to the Graf classification type of hips, 88.71% (3300 patients) had Graf type I normal hips, and 420 children (11.29%) were classified as hip dysplasia or even hip dislocation, as seen in Table 1. In the normal hip group, 51.21% of the children were females and 48.79% of them were males. In the hip dysplasia group, 73.81% were female and 26.19% were male patients (*p* < 0.05). Therefore, we included the 3300 children with normal hips in the Normal Hip group and the 420 children with any type of hip modification in the DDH group.

Considering the order of birth, no correlations were found regarding DDH incidence (*p* = 0.9).

According to the type of birth, C-section delivery was found to be significantly correlated to hip dysplasia, compared to natural delivery (*p* < 0.05).

Regarding baby position at birth, 29.78% (140 patients) of breech presentations and 16.66% (10 patients) of transverse presentations had hip development modifications, compared to only 7.5% (220) of the babies delivered through cephalic presentation (*p* < 0.05).

We considered premature births to be those in which the baby was delivered at least 3 weeks before the due date (before and including 37 weeks of gestation). In the DDH group, 100 patients (23.81%) were prematurely born (*p* < 0.05). In the premature group, 8.19% (50) were diagnosed with DDH, compared to 7.80% (210) of the children born on the due date (*p* = 0.009).

Considering weight at birth, most patients with DDH had a normal weight (88.10%, 370 patients, weighing between 2500 g and 4000 g). Babies weighing below 2500 g were considered to have a low weight for their gestational age. Babies weighing above 4000 g were considered macrosomic. None of the patients that were macrosomic showed hip dysplasia.

None of the twins (130 patients) had DDH.

Oligohydramnios was discovered in 50 (11.9%) pregnancies in the DDH group (*p* < 0.00001). In the normal hip group, 2.42% of pregnancies had oligohydramnios.

Gestational diabetes mellitus was discovered in 2.38% (10 patients) of cases with DDH (*p* = 0.09).

Maternal hypertension during pregnancy was diagnosed in 140 cases, with 30 moms giving birth to 30 children (7.14%) with DDH (*p* = 0.0001).

We considered a scarred uterus to be those cases in which an invasive procedure was carried out on the uterus, such as a previous c-section, curettage, or any other surgical intervention on the uterus, without considering a previous natural delivery. In the DDH group, 30.95% of the patients were born from a scarred uterus (*p* = 0.01).

In the DDH group, 30 patients (7.14%) had a familial history of DDH; either the mother, a sister, or the grandmother was diagnosed with DDH (*p* < 0.00001). One in two children with positive familial history had DDH.

We considered that the child was swaddled if this practice was carried out for at least the first 4 weeks of life during half of the day (mostly at night). A total of 64.29% (270 patients) of patients with DDH were swaddled (*p* < 0.00001).

All of these results are presented in Table 2. Multivariate analysis revealed that gender, prematurity, non-vertex birth presentation, oligohydramnios, gestational diabetes, maternal hypertension, family history, associated deformities, and swaddling are statistically significant risk factors for DDH presence in newborns. Using logistic regression, the generated model had an overall percentage of prediction (accuracy rate) of 91.9%. Considering the omnibus test of model coefficients, the model is significant and has a good fit, compared with the null model (0.000).

About 190 patients (45.24%) from the DDH group had associated deformities such as talipes calcaneovalgus, metatarsus adductus, coxa adducta, congenital clubfoot, torticollis, femur hypoplasia, finger agenesia, and esophagus malformations, as seen in Table 3. We found no statistical significant correlation between clubfoot and DDH in our patient pool.

A total of 3010 children had at least 2 risk factors associated with DDH. Patients from the DDH group are shown in Table 4, along with the number of risk factors. Each one of the seven patients with at least two risk factors had one form of DDH.

## 4. Discussion

DDH is a progressive disease whose prospects and time to diagnosis are inversely proportional. The Global Hip Dysplasia Registry postulates that 1–3 infants per 1000 live births are diagnosed with a dislocated hip, and 20 per 1000 are affected by hip instability. That means an estimated of 2.0–2.6 million infants are affected worldwide each year. Even if a small percentage of the general population develops this condition, late diagnosis and treatment brings a great burden of medical, psychological, and social costs. DDH is the cause of total hip replacement and 28.8% of total hip replacements under 60 years of age [10].

The gold standard tool for diagnosis and monitoring of DDH under 6 months of age is hip ultrasound [11]. It is a technique initially proposed by Graf in the 1980s [10]. The static test assesses the morphology of the hip, while the dynamic test evaluates its stability. According to Graf, it is best performed under 4.5 months of age, while the femoral head is mostly cartilaginous. The recommended age for ultrasound examination in children with positive risk factors is 6 weeks [8].

Laliotis et al. in their cohort study concluded a new ultrasound sign that presents the femoral head and the acetabulum as two concentric circles that are disrupted in infants with DDH. The sign can be reproduced both in the static and dynamic examination of the hip. It can be used in combination with the alpha angle measurement to diagnose the normal development of the infant’s hip [12].

The clinical examination of newborns and instability tests such as Barlow and Ortolani must be carried out in every child, regardless of the risk factors. Moreover, in a child with positive risk factors and a normal clinical and static ultrasound evaluation, it would be indicated that dynamic ultrasound techniques should be performed [13]. If this type of examination also concludes that the hip is normal, a second evaluation must be carried out, both clinical and sonographic, at 4 months to fully exclude an initially occult type of DDH.

There are many papers on the epidemiology and risk factors of DDH in the literature but no standardized criteria on whether a hip ultrasound should be carried out only in selective cases having those risk factors. No international consensus regarding DDH screening has been established. Most North American surgeons do not consider it to be cost-effective to carry out universal ultrasound screening [14], compared to Europe, where countries such as Italy, Austria, Switzerland, Germany, Slovenia, and Slovakia have universal screening programs for DDH [13]. Woodacre et al. proposed calculating an individual risk for each patient, and basing ultrasonography screening on this risk [15].

In 2017, Schams et al. outlined the importance of a universal screening protocol for DDH using the Graf method in all infants [16]. They proved that female gender, breech, and positive family history, as well as the combination of female gender and high weight at birth, are independent risk factors. We conducted screening on every newborn that consulted our team. We noticed that some risk factors were statistically significant for DDH occurrence, such as female gender, prematurity, fetal position at birth, lower birth weight, oligohydramnios, family history of DDH, associated deformities, and swaddling. We found that each factor alone does not influence pathological hip development, but that there is an association of two or more risk factors. On the other hand, patients with normal hips presented none-to-seven risk factors each. Although maternal hypertension and gestational diabetes were statistically significant factors, they cannot be considered true risk factors, as we did not follow all the mothers presenting gestational diabetes or maternal hypertension and their newborns to observe the real impact of those factors on hip development. We found no macrosomic newborn with DDH.

The American Academy of Pediatrics states that an estimated 11.5 infants out of 1000 live births have DDH without any associated risk factor [17]. They also mention a 4-times-higher incidence for girls than for boys and a 1.7-times-higher incidence in patients with positive DDH family history. According to them, breech presentation is a relative risk factor (6.3-times-higher chances of DDH). Dunn indicated that DDH is more frequent in boys with associated deformities and oligohydramnios, while in girls it is induced by maternal hormones that increase capsular laxity [18]. Kokavec et al. mention in their study a sonographic DDH incidence of 69.5 per 1000 children, among which 4.8 needed treatment to recover. The incidence varied from 0.06/1000 in African patients to 76.1/1000 in native Americans, mostly due to genetic inheritance and swaddling customs [19]. In the study of Omeroğlu et al., two-thirds of the infants referred with at least one risk factor and one clinical finding had unilateral or bilateral hip dysplasia [20]. This study concluded that clinical signs of dysplasia are more significant than the coexisting risk factor. Our research showed a 3.2-times-higher incidence for girls than for boys and a higher incidence among patients with a positive family history by a factor of 4.19.

A disadvantage of screening tests is that they cannot be reproduced by all examiners [10]. When performed by trained examiners, clinical screening tests may be cost-effective [10]. On the other hand, a prospective long-term economic analysis in the UK showed that the use of ultrasonography in the diagnosis and surveillance of treatment in DDH does not generate a cost burden, and even reduces the costs for health services and family members later in life [21]. In addition, there is no universally accepted treatment for different stages of dysplasia.

One study has shown that almost 12% of children with DDH remain undiagnosed when selective screening programs are applied [22]. If screening methods were abandoned, then the number of AVN cases would increase, and worse outcomes would be presented because of open reduction surgeries. Ultrasonography carried out by trained examiners will not eliminate the need for surgery because there will still be a number of patients in whom conservative treatment will fail [10].

One particular difficulty we encountered was whether to consider Graf IIa physiological or pathological. Although most of IIa hips resolve spontaneously, and in our opinion they do not need treatment, there are studies that show a risk of up to 10% of negative outcome for IIa hips if left untreated; therefore, we considered them pathological [20,23,24] from a classification point of view. DDH comprises anatomical changes around the hip, and we considered that risk factors are linked to any type of abnormality, even those that may resolve spontaneously. We do not mean to challenge Graf’s classification in a clinical context. Perhaps IIa-hips can have an unfavorable outcome if left untreated. We did not note this in our study pool. In further studies, a clear delineation between IIa- and IIa+ should be made, considering that IIa+ do not need treatment.

Our study’s strength was the large patient pool, with access to the large number of data necessary for a thorough statistical analysis. We conducted a clinical examinations along with ultrasound screening on every child who presented to our ward.

One of our limitations was the lack of long-term follow-up data regarding age dependent treatment and results. Patients with DDH need to be evaluated in long-term studies that span across generations, to clarify the best treatment methods for each stage. Another limitation was that we did not use in our study clinical examination data such as thigh crease asymmetry from the Barlow/Ortolani tests. Our doctors routinely conduct a complete physical examination on the newborns, but these variables were not recorded in the patient history. Clinical findings are important, and further studies should integrate them into DDH assessment and management.

Future research is needed to see whether the threshold of two risk factors is adequate for narrowing the screening population, and also to monitor which treatment is the best, regarding complications and relapse.

## 5. Conclusions

It is mandatory for children with two or more risk factors to be evaluated clinically and sonographically.

Children with clinical signs of DDH should be mandatorily evaluated sonographically.

DDH screening is recommended for each newborn for the long-term benefits of early DDH diagnosis, regardless of the risk factors.

## Figures and Tables

**Figure 1 children-10-00968-f001:**
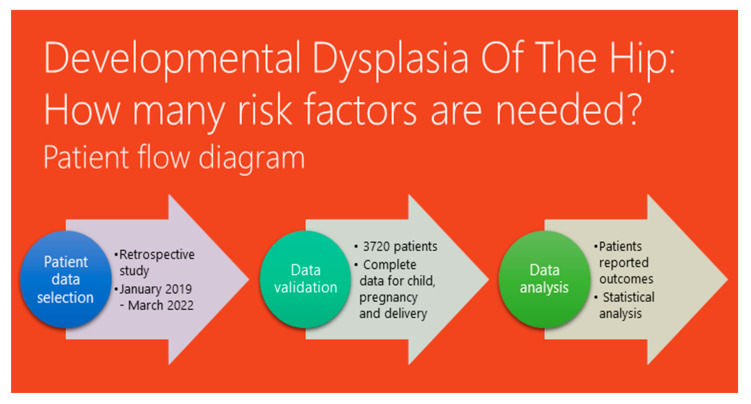
Flow diagram describing the patient selection process from starting until data analysis and patients’ reported outcomes.

**Figure 2 children-10-00968-f002:**
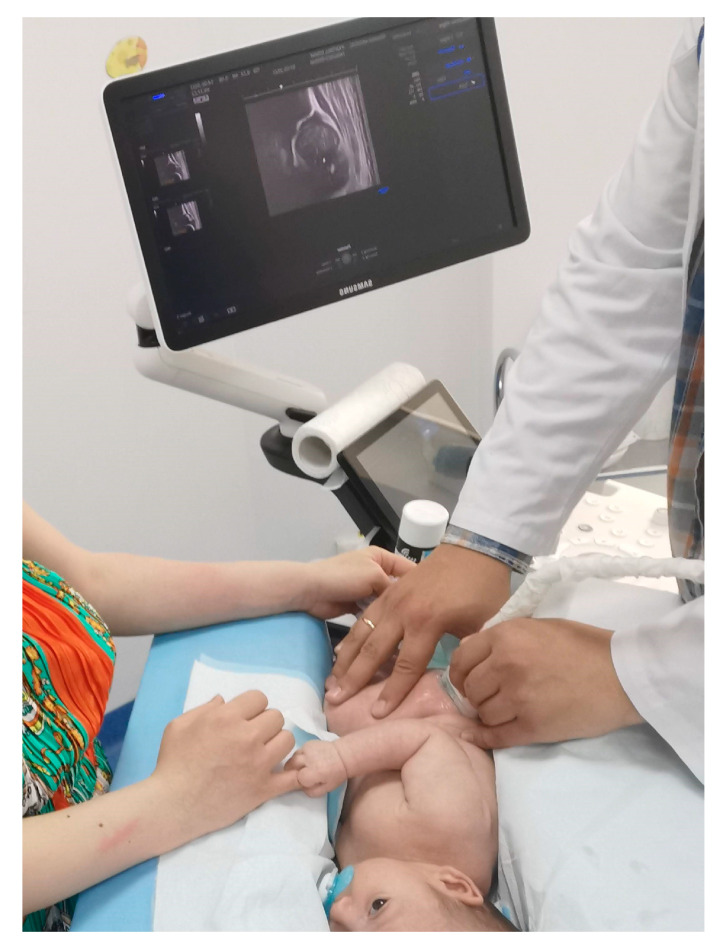
Image depicts the ultrasound procedure for DDH screening in infants. The child is positioned on one side in a specially designed cradle, while the parent holds the upper part in place. The physician conducts the ultrasound examination by manipulating the probe with one hand and stabilizing the lower limb with the other hand. From our collection of medical photos.

**Figure 3 children-10-00968-f003:**
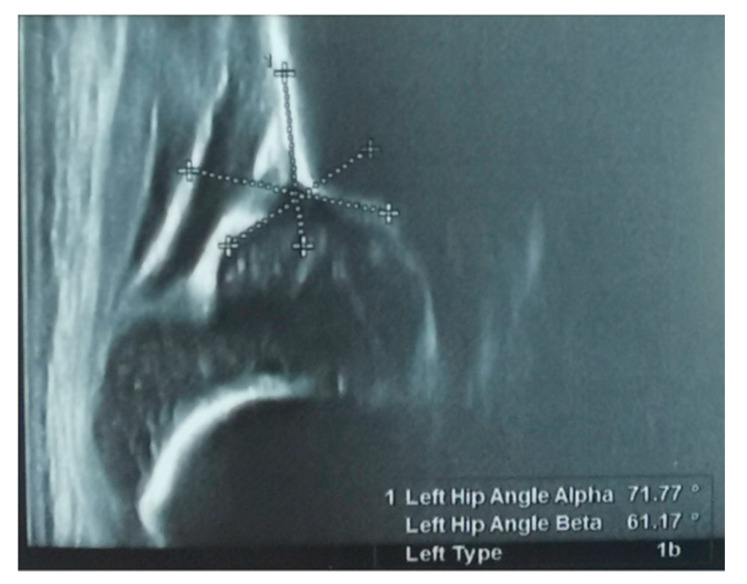
Ultrasound measurements on a left hip ultrasound according to Graf ultrasound technique for the screening of development dysplasia of the hip. Image showing a Graf type 1b hip. From our collection of medical photos.

**Table 1 children-10-00968-t001:** Patients distribution according to Graf Hip Classification and gender. A total of 88.71% of the patients were classified as Graf Hip Type 1, while 11.29% were classified as hip dysplasia.

		Hip Type							Total	*p*
		I	IIa	IIb	IIc	D	III	IV		
	Number of patients	3300	260	10	30	60	40	20	3720	
**Gender**	Female	1690	200	10	20	40	40	0	2000	<0.05
	Male	1610	60	0	10	20	0	20	1720	

**Table 2 children-10-00968-t002:** Risk factors for developmental dysplasia of the hip and Graf hip type classification.

		Hip Type							Total	*p*
		I	IIa	IIb	IIc	D	III	IV		
**Order of birth**	First	2280	170	10	20	40	40	10	2570	0.9
	Other	1020	90	0	10	20	0	10	1150	
**Birth Type**	Natural	440	30	0	0	0	0	0	470	0.0001
	C-section	2860	230	10	30	60	40	20	3250	
**Presentation at birth**	cephalic	2920	220	0	20	20	10	0	3190	0.0001
	breech	330	40	10	10	40	20	20	470	
	transverse	50	0	0	0	0	10	0	60	
**Prematurity**	On due date	2690	210	10	30	30	30	10	3010	0.009
	Premature	610	50	0	0	30	10	10	710	
**Weight at birth**	>4000 g	140	0	0	0	0	0	0	140	n/a
	2500–4000 g	2970	240	10	30	50	30	10	3340	
	<2500 g	190	20	0	0	10	10	10	240	
**Twins**	No	3170	260	10	30	60	40	20	3590	n/a
	Yes	130	0	0	0	0	0	0	130	
**Oligohydramnios**	No	3220	220	10	30	60	40	10	3590	0.0001
	Yes	80	40	0	0	0	0	10	130	
**Gestational diabetes**	No	3120	260	10	20	60	40	20	3530	0.09
	Yes	180	0	0	10	0	0	0	190	
**Maternal hypertension**	No	3190	240	10	20	60	40	20	3580	0.0001
	Yes	110	20	0	10	0	0	0	140	
**Scarred uterus**	No	2470	170	10	20	40	40	10	2760	0.0001
	Yes	830	90	0	10	20	0	10	960	
**Family History**	None	3270	260	10	30	40	30	20	3660	0.00001
	Yes	30	0	0	0	20	10	0	60	
**Swaddling**	No	3060	100	10	10	10	20	0	3210	0.0001
	Yes	240	160	0	20	50	20	20	510	

**Table 3 children-10-00968-t003:** Associated deformities for patients in the study group. Talipes calcaneovalgus and metatarsus adductus were the most common findings.

			Hip Type													Total
			I		IIa		IIb		IIc		D		III		IV	
**Associated Deformities**	None	2230		140		0		20		20		30		20		2460
	Talipes calcaneovalgus	730		80		10		0		20		10		0		850
	Metatarsus adductus	210		40		0		0		0		0		0		250
	Finger agenesia	10		0		0		0		0		0		0		10
	Esophagus malformation	10		0		0		0		0		0		0		10
	Coxa adducta	30		0		0		0		0		0		0		30
	Femur hypoplasia	0		0		0		0		10		0		0		10
	Torticollis	20		0		0		10		0		0		0		30
	Congenital clubfoot	40		0		0		0		10		0		0		50
	Talipes calcaneovalgus and torticolis	20		0		0		0		0		0		0		20
**Total**			3300		260		10		30		60		40		20	3720

**Table 4 children-10-00968-t004:** The number of risk factors associated with Graf hip type for patients presented in the DDH group.

	Hip Type:	IIa	IIb	IIc	D	III	IV	Total
**No. of Risk factors:**	**2**	60		10				**70**
	**3**	90			10	20	10	**130**
	**4**	40	10	10	10	10		**80**
	**5**	60		10	30			**100**
	**6**	10					10	**20**
	**7**	0			10	10		**20**
**Total**		**260**	**10**	**30**	**60**	**40**	**20**	**420**

## Data Availability

Data is available on request.

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
