# Peer review of "Developmental Dysplasia of the Hip: How Many Risk Factors Are Needed?"

_children, 2023, doi:10.3390/children10060968_

Round 1

Reviewer 1 Report

This is an interesting topic, of great importance. The number of patients (over 3000) allows good statistics.

However, I do not agree that IIa hips are pathologic. You can divide them into IIa+ and IIa-, when IIa- can be considered as pathologic. 

IIa group is hips are usually described as physiologic immature. 

So I do not agree to put them in the same "abnormal" hips group together with IIb and III, IV hips. 

Author Response

This is an interesting topic, of great importance. The number of patients (over 3000) allows good statistics.

However, I do not agree that IIa hips are pathologic. You can divide them into IIa+ and IIa-, when IIa- can be considered as pathologic.

IIa group is hips are usually described as physiologic immature.

So I do not agree to put them in the same "abnormal" hips group together with IIb and III, IV hips.

----------

Dear Reviewer,

Thank you for your consideration.

Type IIa is controversial. When we started the study, we also had concerns about whether to consider it pathological or physiological (due to the fact that most of the cases resolve spontaneously).

By looking at the literature, we found several articles that mention Hip Type IIa as abnormal/dysplastic, with an existent risk for a negative outcome (of 5-10%) if left untreated:

- https://doi.org/10.4236/ojped.2020.102025

- https://doi.org/10.1007/s11832-012-0476-1

- https://doi.org/10.4274/balkanmedj.2017.1150

- and others.

Thus, we placed it in the Dysplasia group.

DDH comprises anatomical changes around the hip and we considered that risk factors are linked to any type of abnormality, even those that resolve spontaneously almost always. We do not mean to challenge Graf’s classification in any way, we just used it for our statistical analysis and we mentioned in our graphs the frequency of each type. We still consider risk factors relevant for any type of hip that is not type I.

Regarding your review, we expanded the Discussions on this particular situation. Lines 243-249

Thank you for reviewing our article.

Reviewer 2 Report

Dear authors you have presented an INTERESTING paper, including a GREAT number of babies, you have performed CORRECT and ACCURATE analysis of the risk factors.

 A great number of babies are characterized as Graf IIA, the VAST majority of them are diagnosed as DDH, but will eventually become normal. This is the PROBLEM when diagnosis is based on US, then the incidence is high. Mainly if US exam is performed in the 1st month.

 Authors you have NOT REPORTED AT ALL CLINICAL FINDINGS ( REDUCED ABDUCTION, POSITIVE ORTOLANI). This is the WEAK point, that children with NEGATIVE risk factors are missed, if NOT APPROPRIATELY CLINICALLY EXAMINED.

Your study popluation is not a screening test population, they are babies referred for risk factors or FOR SUSPICIOUS CLINICAL FINDINGS. You must report on this.

In Discussion you mention that when clinical exam in normal and static US is normal, in case of risk factors, you perform dynamic US. Dynamic US will be positive in NEWBORNs due to laxity. Please clarify better this. We have recently described the double concentric circle sign on US, for evaluation of hip stability. Your approach on diagnosis of DDH on US ( Graf IIA) is a major concern for your HIGH percentage of DDH, in badies that otherwise will be considered normal. This is the problem of overtreatment in all series of screening. 

Your study confirms well known risk factors

Please comment that CTEV is NOT related to DDH, your finding is correct and is important for the appropriate evaluation of babies with CTEV

Author Response

Dear authors you have presented an INTERESTING paper, including a GREAT number of babies, you have performed CORRECT and ACCURATE analysis of the risk factors.

RE: Dear Reviewer, Thank you for your consideration.

A great number of babies are characterized as Graf IIA, the VAST majority of them are diagnosed as DDH, but will eventually become normal. This is the PROBLEM when the diagnosis is based on US, then the incidence is high. Mainly if US exam is performed in the 1st month.

RE:

Thank you for your consideration.

Type IIa is controversial. When we started the study, we also had concerns about whether to consider it pathological or physiological (due to the fact that most of the cases resolve spontaneously).

By looking at the literature, we found several articles that mention Hip Type IIa as abnormal/dysplastic, with an existent risk for a negative outcome (of 5-10%) if left untreated:

- https://doi.org/10.4236/ojped.2020.102025

- https://doi.org/10.1007/s11832-012-0476-1

- https://doi.org/10.4274/balkanmedj.2017.1150

- and others.

Thus, we placed it in the Dysplasia group.

In our study, we did not come against the Graf classification, we only assessed the risk factors for each subtype of the Graf classification.

We expanded the Discussions section about this particular situation in lines 243-249.

 Authors you have NOT REPORTED AT ALL CLINICAL FINDINGS ( REDUCED ABDUCTION, POSITIVE ORTOLANI). This is the WEAK point, that children with NEGATIVE risk factors are missed, if NOT APPROPRIATELY CLINICALLY EXAMINED.

RE: We have added in the Discussion section at study limitation. Regarding this issue, we also stated in the Discussion section that clinical findings are important, and further, we recommended in the Conclusions that DDH screening is recommended for each newborn regardless of the risk factors. Lines 255-260.

Your study population is not a screening test population, they are babies referred for risk factors or FOR SUSPICIOUS CLINICAL FINDINGS. You must report on this.

RE: In the Results section, Table 2 shows Risk factors for developmental dysplasia of the hip and patients based on Graf Hip type. The parents consulted the doctors regarding abnormal physical aspects of the lower limbs or regarding risk factors. We added this information in lines 70-73. Such findings as limited abduction, Ortolani/Barlow, skin fold asymmetry, and others were assessed by the clinician but were not recorded in the patient history. We mentioned this in the Study limitations. Lines 255-260

In Discussion you mention that when clinical exam in normal and static US is normal, in case of risk factors, you perform dynamic US. Dynamic US will be positive in NEWBORNs due to laxity. Please clarify better this. We have recently described the double concentric circle sign on US, for evaluation of hip stability. Your approach on diagnosis of DDH on US (Graf IIA) is a major concern for your HIGH percentage of DDH, in babies that otherwise will be considered normal. This is the problem of overtreatment in all series of screenings.

RE: We totally agree with you. Hip Type 2A is a controversial matter as stated above. For example, we do not treat Hip type 2a, we only do further US evaluation to ensure normalization of the hip. We did include IIa in our statistical analysis due to the fact that anatomical modifications around the hip are correlated to the risk factors, and strictly regarding Graf classification they are not considered normal.

We read your article and found this new ultrasound sign to be useful, hence we included it in our Discussion. Lines 182-186.

Your study confirms well-known risk factors

Please comment that CTEV is NOT related to DDH, your finding is correct and is important for the appropriate evaluation of babies with CTEV.

CTEV = congenital talipes equino varus

RE: We have added into Results. Lines 159-160.

Thank you for reviewing our article.

Round 2

Reviewer 1 Report

I know there are some doubts about the IIa hips, but in my opinion they are totsally different (IIa+ and IIa-).

Author Response

Dear Reviewer,

We updated lines 243-252 with Discussions uppon IIa+ and IIa- topic.

Thank you for adding quality to our article.